# Do a Few Weeks Matter? Late Preterm Infants and Breastfeeding Issues

**DOI:** 10.3390/nu11020312

**Published:** 2019-02-01

**Authors:** Beatrice Letizia Crippa, Lorenzo Colombo, Daniela Morniroli, Dario Consonni, Maria Enrica Bettinelli, Irene Spreafico, Giulia Vercesi, Patrizio Sannino, Paola Agnese Mauri, Lidia Zanotta, Annalisa Canziani, Paola Roggero, Laura Plevani, Donatella Bertoli, Stefania Zorzan, Maria Lorella Giannì, Fabio Mosca

**Affiliations:** 1Neonatal Intensive Care Unit, Fondazione IRCCS Cà Granda Ospedale Maggiore Policlinico, via Commenda 12, 20122 Milan, Italy; lorenzo.colombo@mangiagalli.it (L.C.); daniela.morniroli@gmail.com (D.M.); giulia.vercesi@policlinico.mi.it (G.V.); lidia.zanotta@policlinico.mi.it (L.Z.); paola.roggero@unimi.it (P.R.); laura.plevani@mangiagalli.it (L.P.); maria.gianni@unimi.it (M.L.G.); fabio.mosca@mangiagalli.it (F.M.); 2Department of Clinical Sciences and Community Health, University of Milan, Via San Barnaba 8, 20122 Milan, Italy; paola.mauri@unimi.it (P.A.M.); annalisacanziani.ac@gmail.com (A.C.); 3Epidemiology Unit, Fondazione IRCCS Ca’ Granda Ospedale Maggiore Policlinico, Via San Barnaba 8, 20122 Milan, Italy; dario.consonni@unimi.it; 4Mother and Child Unit, Università degli Studi di Milano, ATS Città Metropolitana di Milano, 20122 Milan, Italy; MBettinelli@ats-milano.it; 5Direzione Professioni Sanitarie, Fondazione IRCCS Cà Granda Ospedale Maggiore Policlinico, via Commenda 12, 20122 Milan, Italy; irenespreafico@gmail.com (I.S.); patrizio.sannino@policlinico.mi.it (P.S.); donatella.bertoli@policlinico.mi.it (D.B.); stefania.zorzan@policlinico.mi.it (S.Z.); 6Dipartimento Donna-Bambino-Neonato, Fondazione IRCCS Cà Granda Ospedale Maggiore Policlinico, via Commenda 12, 20122 Milan, Italy

**Keywords:** breastfeeding, late preterm, protective factors, promotion of breastfeeding, breastfeeding support

## Abstract

The late preterm infant population is increasing globally. Many studies show that late preterm infants are at risk of experiencing challenges common to premature babies, with breastfeeding issues being one of the most common. In this study, we investigated factors and variables that could interfere with breastfeeding initiation and duration in this population. We conducted a prospective observational study, in which we administered questionnaires on breastfeeding variables and habits to mothers of late preterm infants who were delivered in the well-baby nursery of our hospital and followed up for three months after delivery. We enrolled 149 mothers and 189 neonates, including 40 pairs of twins. Our findings showed that late preterm infants had a low rate of breastfeeding initiation and early breastfeeding discontinuation at 15, 40 and 90 days of life. The mothers with higher educational levels and previous positive breastfeeding experience had a longer breastfeeding duration. The negative factors for breastfeeding were the following: Advanced maternal age, Italian ethnicity, the feeling of reduced milk supply and having twins. This study underlines the importance of considering these variables in the promotion and protection of breastfeeding in this vulnerable population, thus offering mothers tailored support.

## 1. Introduction

Late preterms are defined as infants born from 34 0/7 to 36 6/7 weeks of gestational age (GA) and comprise approximately 75% of all preterm births [1]. Late preterm births are increasing overall due to the increased use of reproductive technologies and, therefore, the occurrence of multiple pregnancies, in combination with advances in obstetric surveillance and interventions, which could lead to the choice of preterm delivery for babies at risk of perinatal complications (intrauterine growth restriction, placental insufficiency or monochorionic multiple pregnancies) [2,3]. Compared with infants born at term, late preterm infants are at increased risk of neonatal morbidity (i.e., hypoglycemia, hypothermia, jaundice, delayed oral feeding, readmission to the hospital, transient tachypnoea, neuro-developmental delays and mortality) [2]. This population has immature organs and systems, including the brain; a compromised immunomodulatory response; and an increased susceptibility to inflammatory injury and oxidative stress [4,5]. For this reason, breast milk with its bioactive components and its combination of nutritional, anti-infective, anti-inflammatory, anti-oxidative, epigenetic, and gut-colonizing substances is particularly important for these infants [4,5,6,7].

Despite these benefits, late preterm infants have a decreased likelihood of breastfeeding initiation and shortened breastfeeding duration compared to term newborns [2,4,8]. To date, few studies have investigated the modifiable factors affecting breastfeeding and discussed the best approaches to promoting and supporting breastfeeding in this vulnerable population [2,4,8]. Kair et al. determined barriers and facilitators of breastfeeding continuation among a large cohort of late preterm infants and compared preterm infants admitted to the well-baby nursery to those admitted to the neonatal intensive care unit (NICU), concluding that NICU admission was not associated with early breastfeeding cessation [6]. Moreover, the experience of breastfeeding late preterm infants was investigated in a cohort of 44 mothers who were interviewed by phone up to 12 months after delivery [9]. Reduced milk supply, difficulties in latching, feelings of failure, and inadequate lactation support from health care providers after discharge have been identified as the most challenging difficulties encountered by the mothers [9].

The purpose of our research is to investigate the variables that could affect breastfeeding duration in a population of late preterm infants admitted to a well-baby nursery and followed up for three months after birth.

## 2. Materials and Methods

A prospective, observational, single-center study was carried out in the well-baby nursery of Fondazione IRCCS Ca’ Granda Ospedale Maggiore Policlinico in Milan, Italy. The Ethics Committee of the “Fondazione Istituto di Ricovero e Cura a Carattere Scientifico Cà Granda Ospedale Maggiore Policlinico” endorsed the present study, and parents provided written informed consent.

We enrolled all consenting mothers with an adequate comprehension of the Italian language who had given birth to late preterm infants admitted to the well-baby nursery from October to December 2017. According to our internal protocol, late preterm infants with a GA from 34 0/7 to 36 6/7 weeks (estimated based on the last menstrual period) and birth weight ≥ 1800 g were admitted to the well-baby nursery, provided that their clinical conditions were stable. This means no need of intravenous infusions, noninvasive respiratory support, or any type of invasive assistance, whereas monitoring of vital parameters and/or the need of incubator were possible in the nursery in the first 24 h of life. We did not enroll a control group of term neonates. The exclusion criteria were the following: Hospitalization in the NICU, congenital anomalies, genetic syndromes, respiratory diseases, neurologic problems, metabolic disorders, or gastrointestinal problems. Before discharge, a structured questionnaire for each infant, including twins, was administered to mothers by one of five health care professionals. The questionnaire used was created by the public health organization of the city of Milan (ATS Città di Milano) for monitoring and comparing breastfeeding habits during the Baby-Friendly Hospital Initiative (BFHI) accreditation [10] of the city’s hospitals. It included closed-ended questions and took fifteen minutes to administer.

The questionnaire assessed sociodemographic characteristics (maternal age, education, ethnicity), basic characteristics of newborns (GA, birth weight), previous experiences (participation in a prenatal class and previous experience with breastfeeding), type of delivery, peripartum experiences (skin-to-skin contact for at least two hours after birth and rooming-in, defined as the baby being kept in the mother’s room for at least 23 h a day), factors affecting lactation (latching difficulties; use of a pacifier; the feeling of reduced milk supply, defined as a subjective maternal perception of having not enough milk to satisfy the baby; breastfeeding on demand or scheduled breastfeeding), and mode of feeding.

Our breastfeeding policy was based on the principles of the BFHI [10]. For this reason, the process of breastfeeding on demand was always explained to mothers and actively promoted by all health care professionals. In contrast, scheduled feeding, defined as feeding the baby at set intervals, was always the mother’s choice. The mode of breastfeeding was reported according to the World Health Organization (WHO) definitions [11]: “Exclusive breastfeeding” indicates no food or drink other than breast milk; and “predominant breastfeeding” indicates that the infant’s predominant source of nourishment was breast milk but that the infant also received other liquids (i.e., water and water-based drinks). “Mixed feeding” refers to the use of both breast milk and formula.

We used obstetric charts and infants’ computerized medical charts (Neocare, i & t Informatica e Tecnologia Srl, Italy) to collect the basic characteristics of the newborns (i.e., birth weight and GA), mode of breastfeeding, and accurate reports of the skin-to-skin contact time and rooming-in time. All the other data were obtained from the questionnaires.

The mothers completed the first follow-up questionnaire during the visit after discharge, within the first seven days of their infants’ lives. Then, structured phone interviews were conducted at 15, 40, and 90 days after birth. The data on the variables subject to changes over time (i.e., latching difficulties, use of a pacifier, mode of feeding) were also collected during each follow-up period. These interviews lasted approximately fifteen minutes. The health care providers in charge of collecting the data during the hospitalization were the same providers responsible for the administration of the questionnaires and phone interviews. With regard to twins, mothers completed the first follow-up questionnaire and the subsequent phone interviews separately for each infant.

To take into account intra-individual correlations over time and the probability of exclusive breastfeeding according to selected variables (including those related to sociodemographic features, clinical characteristics, delivery, peripartum, and past or current lactation determinants), odds ratios (ORs) and 95% confidence intervals (CIs) were calculated with univariate and multiple generalized estimation equation (GEE) logistic regression models. The statistical analyses were performed using Stata 15 (StataCorp LLC, College Station, TX, USA, 2017).

## 3. Results

The total eligible population included 149 mothers and 189 neonates, including 40 pairs of twins. Seventeen percent of these neonates (32 neonates) were born at 34 weeks of GA, 33.3% (63 neonates) were born at 35 weeks of GA, and 49.7% (94 neonates) were born at 36 weeks of GA. Neonatal birth weight ranged from 2050 g to 2990 g. Only 119 mothers completed the study, with a final dropout rate of 20%. The basic characteristics of the mothers are shown in Table 1.

The minimum maternal age was 20 years, and the maximum age was 49 years. Italians represented 82.5% of the study population: Among them, 56.8% had a degree and 51.2% attended a prenatal class. However, in the population of foreign mothers, there was a lower educational level (30.8% had a degree and 34.6% attended a prenatal class). In contrast to the Italian mothers, foreign mothers were more likely to breastfeed on demand (27% vs. 18%) and to have had positive breastfeeding experiences (57.7% vs. 45.5%). Most of the population did not experience skin-to-skin contact (95.5%) or rooming-in (88.6%).

Table 2 shows the proportion of different types of feeding over the first three months of life.

At discharge, only 16.8% of mothers exclusively breastfed their babies, and the rate increased to a maximum of 40.3% at 15 days of life. Mixed feeding peaked at discharge (78.8%) and decreased over time, while the rate of formula feeding progressively increased. Among the lactation factors also reported in Table 2, incorrect latching of the baby on the breast (even if only reported by the mother) was more common during the first days after birth, while the perception of reduced milk supply occurred later.

For each investigated variable, no difference was found in the answers provided by the mothers with regard to twins.

According to univariate and multivariate regression (Table 3), the factors found to be protective for exclusive breastfeeding were a previous positive experience of breastfeeding and maternal education. Maternal age over 35 years, Italian ethnicity, the feeling of reduced milk supply, and twin pregnancies were risk factors for the early cessation of breastfeeding.

## 4. Discussion

Late preterms are increasing in number overall, adding substantially to the impact on health care services, both in the acute, and primary health care settings [2,12]. This study contributed to expanding the current body of knowledge by increasing awareness of how to best target support for mothers of late preterm infants.

Our data confirm that compared to infants born at term in the same hospital setting and in a similar sociodemographic context, late preterm infants were less likely to be exclusively breastfed in the hospital and after discharge, and required more time to acquire breastfeeding skills [13]. Accordingly, the breastfeeding rate in the late preterm population progressively increased to 40.3% at 15 days of life, whereas Colombo et al. reported the peak rate of exclusive breastfeeding to be 75.3% in term newborns at discharge [13]. The present study identified an increased maternal age, Italian ethnicity, low maternal education, lack of previous positive breastfeeding experiences, having twins, and the feeling of reduced milk supply as risk factors for the early discontinuation of exclusive breastfeeding.

Little is known about the effect of maternal age on breastfeeding rates for late preterm infants, and the results are conflicting when different social contexts are considered [14]. Nevertheless, the most developed regions have seen an increase in maternal age in the last two decades [15], and our data indicate that mothers aged 35 years or older are less likely to breastfeed. Accordingly, increased maternal age has been associated with a negative effect on breastfeeding among mothers of term infants in the same hospital setting with a similar sociodemographic context [13].

Italian mothers were less likely to breastfeed than were foreign mothers, unlike what was observed in a population of mothers of term newborns [13]. It can be hypothesized that the limited number of foreign mothers and the high percentage of Italian mothers older than 35 years could have affected this result.

Regarding the mode of delivery, the literature has shown that cesarean is negatively associated with breastfeeding [2,16,17]. Studies established a negative association between cesarean delivery, the onset of lactation, milk transfer and milk production [2,16,17]. Accordingly, Ayton et al. reported that late preterm infants delivered by cesarean section were 80% less likely to initiate breastfeeding within one hour of birth and identified the combination of cesarean delivery with prematurity as a synergic risk factor that creates a complex breastfeeding scenario [2]. In the univariate analysis, our data confirmed the known negative association between cesarean section and breastfeeding. However, the significance was lost in multivariate analysis, probably because of the influence of other variables; increased maternal age and/or having had an iterative elective cesarean section could have implied a previous maternal breastfeeding experience.

Rooming-in and skin-to-skin contact at least two hours after birth are well-known factors associated with successful breastfeeding [18]. Skin-to-skin contact encourages breastfeeding behavior, stimulates innate reflexes in newborns [19], accelerates neurophysiological development [20], and is positively associated with breastfeeding duration in preterm infants [19]. However, in our study, we could not find any significant associations among skin-to-skin contact, rooming-in and breastfeeding, probably due to the very small percentage of mothers who experienced one or both of these practices. The limited implementation of skin-to-skin contact and the partial implementation of rooming-in in our population may be attributable to the fact that late preterm infants frequently need an incubator for the first hours or possibly days of life.

The use of pacifiers could be justifiable in certain situations, particularly in a NICU setting and when the neonate is separated from the mother. The pacifier could support breastfeeding in low-birth-weight and premature infants by promoting the maintenance and maturation of the sucking reflex; it also helps to achieve a neurobehavioral organization and could relieve pain and decrease stress [21]. Other studies have reported that, both in term and preterm infants, the minimization of the use of a pacifier positively affected exclusive breastfeeding [13,19]. However, in our late preterm population in a well-baby nursery setting, avoiding the use of pacifiers was not found to be associated with breastfeeding.

Twin pregnancies were strongly correlated with the early cessation of breastfeeding in our population. Our possible explanation could be the greater maternal fatigue in simultaneously managing and feeding more than one premature infant. Few studies examining breastfeeding among preterm infants include or differentiate among multiple pregnancies. As reported by Giannì et al., having twins was reported by the mother’s of late preterm infants to be a barrier to breastfeeding [8]. In contrast, Demirci et al. observed a higher rate of breastfeeding initiation among mothers of late preterm multiples than among mothers of singleton infants [14].

Maternal education was positively associated with breastfeeding. This finding was not surprising, and it was likewise previously described as a predictor of a longer duration of breastfeeding in both term [13] and late preterm infants [6,22]. With regard to lactation factors, the feeling of reduced milk supply and latching difficulties negatively affected breastfeeding, while breastfeeding on demand was strongly associated with the continuation of breastfeeding, according to the literature [6,23]. The maternal perception of inadequate milk supply, breastfeeding difficulties, and concerns that breast milk alone did not satisfy the infants were reported as the top three reasons for breastfeeding discontinuation in the study of Kair et al., which surveyed a total of 2530 mothers of late preterm infants [6]. Moreover, the same barriers were reported in two qualitative [9,24] studies that illustrated a common phenomenon of maternal frustration associated with breastfeeding difficulties.

Breastfeeding a late preterm infant could be challenging due to the peculiar characteristics of this population, such as relative hypotonia and consequent ineffective sucking [2] and immaturity in coordinating swallowing and breathing during breastfeeding [25]. In this scenario, the availability of expert lactation support is crucial [8,26] and affects mothers’ self-efficacy, which could be crucial in the establishment of breastfeeding, even in late preterm populations [27,28]. We also believe that mothers’ self-efficacy may be correlated with personal experiences. Previous encouraging breastfeeding experiences were strongly positively associated with breastfeeding initiation and duration, whereas a previous negative breastfeeding experience affected breastfeeding initiation and duration much more than having no experience at all. These findings are consistent with those concerning term infants [13] and reinforce the need to closely explore mothers’ backgrounds to identify any possible breastfeeding issues and specific groups of mothers that will need extra help.

The present study has some limitations. First, the data were collected from a single institution where internal procedures are applied and thus it is not possible to generalize our findings. Moreover, the results were not stratified according to gestational age. However, it addressed a relatively large number of mothers who received the same modalities of support, and all infants were late preterm infants without comorbidities and with a low-risk birthweight, making the population homogenous and composed of infants with other risk factors, except being born as a late preterm.

## 5. Conclusions

Many studies have demonstrated that, even when considered healthy, this population is at risk of having issues typical of premature babies, with breastfeeding problems being the most predictable. Considering the importance of mothers’ milk and its beneficial effects for both term and preterm infants, the main significance of our study was to underline the need for all health care professionals to be aware of the support required for breastfeeding late preterm infants. The implementation of specific and timely breastfeeding support measures for mothers (especially if old, having twins, with low education, with no previous positive breastfeeding experiences, or with the perception of reduced milk supply), could facilitate to give their preterm infants the best start in life.

## Figures and Tables

**Table 1 nutrients-11-00312-t001:** Population features.

	Foreign Mothers (*n* = 26)		Italian Mothers (*n* = 123)		Total (*n* = 149)	
	*n*	%	*n*	%	*n*	%
**Sociodemographic features**						
**Maternal age**						
20–29 years	8	30.8	8	6.5	16	10.7
30–34 years	10	38.5	37	30.1	47	31.5
35–39 years	5	19.2	41	33.3	46	30.9
40–49 years	3	11.5	37	30.1	40	26.8
**Education**						
High school diploma	18	69.2	53	43.1	71	47.7
Degree	8	30.8	70	56.9	78	52.3
**Previous experiences**						
**Prenatal classes**						
Yes	9	34.6	63	51.2	72	48.3
No	17	65.4	60	48.8	77	51.7
**Previous breastfeeding experience**						
Positive	7	27	22	18	29	19.4
None	19	73	88	71.5	107	71.8
Negative	0	0	13	10.5	13	8.8
**Delivery and peripartum experiences**						
**Mode of delivery**						
Eutocic	8	30.8	35	28.4	43	28.9
Vacuum/forceps	0	0	6	4.9	6	4.0
Emergency cesarean section	8	30.8	36	29.3	44	29.5
Elective cesarean section	10	38.4	46	37.4	56	37.6
**Twins**	8	20	32	80	40	100
**Skin-to-skin contact**						
Yes	2	7.7	5	4.1	7	4.7
No	24	92.3	118	95.9	142	95.3
**Rooming-in**						
Yes	5	19.2	12	9.8	17	11.4
No	21	80.1	111	90.2	132	88.6
**Lactation factors**						
**Type of breastfeeding**						
Scheduled	11	42.3	67	54.5	78	52.3
On demand	15	57.7	56	45.5	71	47.7

**Table 2 nutrients-11-00312-t002:** Follow-up findings.

Variable	Before Discharge * (*n* = 149)	First Visit after Discharge (*n* = 138)	15 Days of Life (*n* = 134)	40 Days of Life (*n* = 127)	90 Days of Life (*n* = 119)
	*n* (%)	*n* (%)	*n* (%)	*n* (%)	*n* (%)
**Lactation factors**					
Latching difficulty	44 (29.5)	29 (21)	28 (20.9)	13 (10.2)	8 (6.7)
Pacifier	42 (28.1)	45 (32.6)	51 (38)	69 (54.3)	72 (60.5)
Feeling of reduced milk supply	5 (3.3)	6 (4.4)	10 (7.4)	12 (9.4)	5 (4.2)
**Type of feeding**					
Exclusive breastfeeding	25 (16.8)	53 (38.4)	54 (40.3)	43 (33.8)	37 (31.1)
Predominant breastfeeding	0 (0.0)	0 (0.0)	0 (0.0)	2 (1.7)	6 (5.0)
Mixed feeding	113 (75.8)	71 (51.5)	64 (47.8)	46 (36.2)	25 (21.0)
Formula feeding	11 (7.4)	14 (10.1)	16 (11.9)	36 (28.3)	51 (42.9)
Lost to follow up	0	11(7.3)	15 (10)	22 (14.8)	30 (20)

* “Before discharge” refers to the 24 h period before discharge.

**Table 3 nutrients-11-00312-t003:** Factors affecting breastfeeding: results of univariate and multivariate logistic regressions.

	OR Crude (95% CI)	OR Adjusted * (95% CI)
**Sociodemographic features**		
**Maternal age**		
Maternal age < 30 years	1.0 (reference)	1.0 (reference)
Maternal age 30–34 years	1.5 (0.6–3.9)	0.8 (0.2–2.9)
Maternal age 35–39 years	0.5 (0.2–1.4)	0.3 (0.1–0.9)
Maternal age > 40 years	0.3 (0.1–0.8)	0.2 (0.0–0.7)
**Education**		
High school diploma	1.0 (reference)	1.0 (reference)
Degree	1.2 (0.6–2.1)	2.2 (1.0–5.0)
**Ethnicity**		
Foreign mothers	1.0 (reference)	1.0 (reference)
Italian ethnicity	0.4 (0.2–0.7)	0.2 (0.0–0.7)
**Previous experiences**		
**Prenatal classes**		
No prenatal classes	1.0 (reference)	1.0 (reference)
Prenatal classes	1.5 (0.8–2.9)	1.5 (0.7–3.3)
**Clinical history**		
No breastfeeding experience	1.0 (reference)	1.0 (reference)
Previous negative experience with breastfeeding	0.7 (0.2–2.2)	0.9 (0.2–2.8)
Previous positive experience with breastfeeding	3.1 (1.4–7.0)	3.0 (1.0–8.5)
**Delivery and peripartum experiences**		
**Mode of delivery**		
Eutocic delivery	1.0 (reference)	1.0 (reference)
Vacuum assisted delivery	2.4 (0.5–11.5)	1.9 (0.4–10.9)
Emergency cesarean delivery	0.4 (0.2–0.9)	0.8 (0.3–2.5)
Elective cesarean delivery	0.3 (0.2–0.8)	0.5 (0.2–1.5)
**Twins**		
Singleton	1.0 (reference)	1.0 (reference)
Twins	0.2 (0.1–0.4)	0.2 (0.1–0.5)
**Skin-to-skin contact**		
No skin-to-skin contact	1.0 (reference)	1.0 (reference)
Skin to skin contact	3.2 (0.7–15.2)	9.9 (0.8–119.0)
**Rooming-in**		
No rooming-in	1.0 (reference)	1.0 (reference)
Rooming-in	1.1 (0.4–3.1)	0.7 (0.2–2.6)
**Lactation Factors**		
**Type of breastfeeding**		
Scheduled breastfeeding	1.0 (reference)	1.0 (reference)
On demand	1.8 (0.9–3.3)	1.4 (0.6–2.9)
**Latching**		
No difficulties	1.0 (reference)	1.0 (reference)
Attachment difficulties	0.8 (0.5–1.3)	0.8 (0.5–1.2)
**Pacifier**		
Use of pacifier	1.0 (reference)	1.0 (reference)
No use of pacifiers	1.3 (0.9–1.7)	1.2 (0.7–2.0)
**Milk supply**		
No reduced milk supply	1.0 (reference)	1.0 (reference)
Feeling of reduced milk supply	0.4 (0.2–0.7)	0.3 (0.1–0.5)

* Each factor was adjusted for the others. Odds ratios were from univariate and multiple generalized estimation equation logistic regression models. OR: odds ratio; CI: confidence interval.

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
