# Peer review of "Do a Few Weeks Matter? Late Preterm Infants and Breastfeeding Issues"

_nutrients, 2019, doi:10.3390/nu11020312_

Round 1
Reviewer 1 Report
Thank you for the opportunity to review this interesting paper. I have compiled a list of comments which I've attached in a separate file.

Author Response
Dear Reviewer,
We appreciate your suggestions and your comments. As you suggest, we edit English language and style and we enclose the certificate. On the following, the replies to your comments are listed point-by-point.
Sincerely,
The Authors

Reviewer 2 Report
This is a good and accurate review of the literature enhanced by new data, which supports previous reports. The new data would have been stronger if the number of subjects (babies and mothers) had been increased by at least 100 percent. The conclusions and recommendations are sound and in agreement with what is generally recommended. The English writing is in need of significant editing for style, spelling, grammar and clarity, although the message is clearly presented.
Author Response
January 18, 2019
Dear Reviewer,
We appreciate your suggestions and your comments. As you suggest, we did extensive editing of English language and style and we enclose the certificate.
Sincerely,
The Authors

Round 2
Reviewer 1 Report
Thank you for giving me the opportunity to read your interesting manuscript. You have addressed my comments from the previous review satisfactorily.
Author Response
Thank you for having reviewed our article.
We previously replied to comments and suggestions.